# MUSICAL SCORE UNDERSTANDING BENCHMARK: EVALUATING LARGE LANGUAGE MODELS' COMPREHENSION OF COMPLETE MUSICAL SCORES

## ABSTRACT

Understanding complete musical scores requires reasoning over symbolic structures such as pitch, rhythm, harmony, and form. Despite the rapid progress of Large Language Models (LLMs) and Vision-Language Models (VLLMs) in natural language and multimodal tasks, their ability to comprehend musical notation remains underexplored. We introduce Musical Score Understanding Benchmark (MSU-Bench), the first large-scale, human-curated benchmark for evaluating score-level musical understanding across both textual (ABC notation) and visual (PDF) modalities. MSU-Bench comprises 1,800 generative question-answer (QA) pairs drawn from works spanning Bach, Beethoven, Chopin, Debussy, and others, organised into four progressive levels of comprehension: Onset Information, Notation & Note, Chord & Harmony, and Texture & Form. Through extensive zero-shot and fine-tuned evaluations of over 15+ state-of-the-art (SOTA) models, we reveal sharp modality gaps, fragile level-wise success rates, and the difficulty of sustaining multilevel correctness. Low-Rank Adaptation (LoRA) markedly improves performance in both modalities while preserving general knowledge, establishing MSU-Bench as a rigorous foundation for future research at the intersection of AI, musicological, and multimodal reasoning.

## 1 INTRODUCTION

Large Language Models (LLMs) and Vision-Language Models (VLLMs) have demonstrated exceptional capabilities in understanding and generating human language, leading to significant progress in a wide range of Natural Language Processing (NLP) tasks (Brown et al., 2020; Chowdhery et al., 2022; OpenAI, a;b). However, their capacity to reason about complete musical scores remains largely unexplored. Existing benchmarks (Yue et al., 2024a; Chen et al., 2025; Li et al., 2024; Yuan et al., 2024; Wang et al., 2025b) for musical score comprehension are limited in scope, as they typically focus on isolated fragments, short excerpts, or multiple-choice tasks rather than fostering a holistic understanding of entire scores. Furthermore, studies such as (Yuan et al., 2024; Wang et al., 2025b) address mainly monophonic music, which consists of a single melodic line without harmonic or rhythmic accompaniment. These approaches are insufficient for capturing the complexity and richness required for open-ended, real-world musicological reasoning.

In complete scores, VLLMs face two persistent challenges. The first is localisation: models often fail to correctly identify bar positions, a prerequisite for answering higher-level musicological questions concerning harmony, texture, or form. For example, when asked "Which articulation is used in bar 7?", the model misaligns the bar and outputs incorrect markings (see Figure 1a). The second challenge is hallucination, where models fabricate content not grounded in the score, often compounding errors from bar mislocalisation. This leads to unreliable interpretations of complete scores, undermining trust in model outputs when compared with the ideal answer (see Figure 1b).

We empirically show that these issues can be mitigated by representing complete scores in ABC notation (Ma et al., 2024). ABC notation is a text-based symbolic format that encodes bar position, pitch, rhythm, and articulation using human-readable characters, thereby providing a structured representation that is readily interpretable by LLMs. An example of metadata and musical content encoded in ABC notation is shown in Figure 2b and Figure 2c.

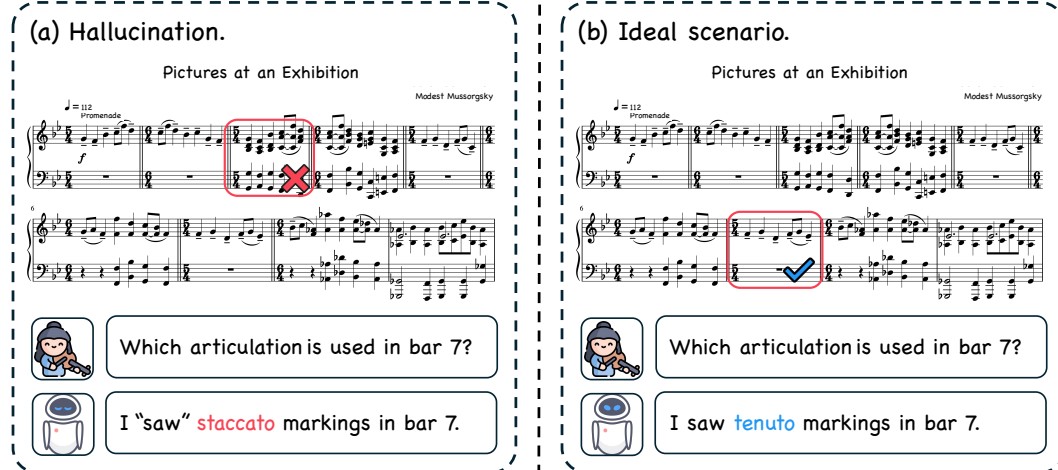

Figure 1: (a) **Hallucination.** When queried about specific score features in bars, VLLMs often fabricate responses that are not grounded in the actual score. (b) **Ideal scenario.** Models should accurately localise and analyse bars, thereby supporting reliable higher-level musicological reasoning.

To evaluate the capacity to reason about complete musical scores, our principal contributions are as follows: (1) We introduce MSU-Bench, the first large-scale benchmark for evaluating LLMs and VLLMs on complete musical scores, comprising 1,800 human-curated generative QA pairs across four progressive levels, spanning four hierarchical levels of musical comprehension: **(1) Onset Information**, **(2) Notation & Note**, **(3) Chord & Harmony**, and **(4) Texture & Form**; (2) it enables multimodal evaluation through textual QA in ABC notation and visual QA in PDF scores; (3) zero-shot experiments on 15+ SOTA models reveal a pronounced textual–visual gap, fragile level-wise success rates, and limited robustness across levels; (4) LoRA (Hu et al., 2021) achieves substantial improvements in both modalities while retaining general knowledge; (5) asking questions one by one yields better performance than all at once, suggesting that hierarchical scaffolding may not be effectively leveraged by current models.

## 2 RELATED WORK

**Musical Score Representation.** Musical score understanding constitutes a key task of Music Information Retrieval (MIR), aiming to analyse and interpret symbolic music representations in order to support downstream applications such as genre and style recognition (Simonetta et al., 2019). Drawing on approaches in representation learning, earlier studies have frequently employed Optical Music Recognition (OMR) to convert scores into digital formats, such as MIDI (Moore, 1988), MusicXML (Good et al., 2001), and LilyPond (Nienhuys & Nieuwenhuizen, 2003), thereby facilitating the learning of embeddings that capture musical structure and semantics for these understanding tasks (Zeng et al., 2021; Liang et al., 2020; Chou et al., 2021). On the other hand, musical notation systems, such as ABC notation, encode musical elements using an alphabetic system with ASCII characters (Gorn et al., 1963). Its concise, high-compression, and language-compatible format makes it particularly suited for integration with large language models, enabling symbolic music understanding and generation (Tang et al., 2025; Wang et al., 2025a).

**QA Benchmarks for Score Understanding.** Currently, the research area has shown increasing interest in QA tasks for score understanding, which require more advanced forms of musical comprehension (Yue et al., 2024b). Notably, MusicTheoryBench (Yuan et al., 2024) represents a systematic attempt to assess the competence of LLMs in music theory, evaluating performance across tasks that demand both music knowledge and reasoning. MusiXQA (Chen et al., 2025) evaluates VLLMs in their ability to interpret musical scores represented as images. ZIQI-EVal (Li et al., 2024) benchmarks LLMs on tasks of music comprehension and generation, with particular emphasis on their capacity to integrate contextual and cultural background knowledge. Furthermore, SSMR-Bench (Wang et al., 2025b) introduces a synthetic data generation framework capable of producing both textual and visual question formats to support comprehensive evaluations of musical understanding.

## 3 BENCHMARK DESIGN

### 3.1 RESEARCH QUESTIONS

MSU-Bench aims to inspire future research in the field of musical score understanding using LLMs and VLLMs, and particularly, it seeks to investigate the following Research Questions (RQs):

**RQ1: How accurately can a model identify onset-level musical metadata?**

**Level 1 (Onset Information).** Level 1 questions assess whether a model can accurately extract onset-level musical metadata from symbolic scores. This information constitutes the foundation for more advanced analysis and performance. Critical aspects include identity-related details such as composer, title, and instrumentation; notational elements including key signature, clef, and time signature; performance onset indicators such as tempo, metronome markings, and expressive or dynamic instructions; and initial structural features of the score, for instance, the presence of an anacrusis. Collectively, these elements provide essential information which is necessary for evaluating a model's ability to interpret advanced musical information.

**RQ2: How correctly can a model interpret local notational and pitch-level features?**

**Level 2 (Notation & Note).** This level focuses on note-to-note and bar-level notation, rather than on global metadata (**RQ1**). It highlights the capacity to identify localised score features that are crucial for understanding musical texture and performance detail. Central questions concern the identification of pitch range, accidentals, rests, ornaments, articulations, dynamics, clef, key and time signature changes, tempo changes, and repeat signs within a given bar or group of bars.

**RQ3: To what extent can a model accurately analyse harmonic structures in symbolic scores?**

**Level 3 (Chord & Harmony).** Unlike **RQ1** and **RQ2**, which focus on onset-level metadata and local notational features, level 3 moves beyond surface description to address the higher-order organisation of harmony. It focuses on the recognition of chord qualities and functions (major, minor, seventh, diminished), together with structural features such as inversions, voicing, spacing, and the treatment of omitted or repeated notes. It also addresses the interpretation of chord progressions across multiple bars, including considerations of whether a piece begins on the tonic and how tonal stability is sustained. In addition, this level encompasses the identification of cadential patterns (perfect, imperfect, interrupted, auxiliary), the presence of dominant or tonic pedals, and ornamental harmonic devices such as suspensions and anticipations. Finally, it involves tracing key and tonal changes, from the initial state through mid-piece modulations to the eventual reassertion of the tonic.

**RQ4: To what extent can a model analyse textural and formal aspects of musical works?**

**Level 4 (Texture & Form).** Level 4 extends the scope of **RQ3** to global dimensions of texture and form, addressing how musical materials are structured, developed, and distributed across the entire work. This level of investigation examines a model's capacity to analyse the textural and formal dimensions of musical works. It entails recognising and interpreting melodic motifs, such as their characteristics, placement, variation, and development, together with the organisation of principal and secondary themes and transitional passages. It also involves identifying textural and structural features such as accompaniment types, vocal or instrumental scoring, and orchestration, as well as broader formal categories including genre, form, and performance medium. Finally, it requires sensitivity to registral distribution, considering how melodic material is allocated across bars, voices, or instruments within the score.

### 3.2 CASE STUDY

We present a case study to illustrate the structure of Levels 1–4 questions in MSU-Bench, demonstrating that ABC notation supports musical understanding rather than serving solely as a textual representation of the score in Figure 2a. ABC notation consists of two principal components: metadata (Figure 2b) and musical content (Figure 2c). As shown in Figure 2, the ABC notation encodes both structural and performance details of Mussorgsky's *Pictures at an Exhibition*, while also providing sufficient symbolic information to address questions across all four levels (see Figure 2d).

**Metadata Information.** The ABC header begins with `X:1`, which identifies this as tune number one in the file. The title of the piece is given as `T:Pictures at an Exhibition`, and the

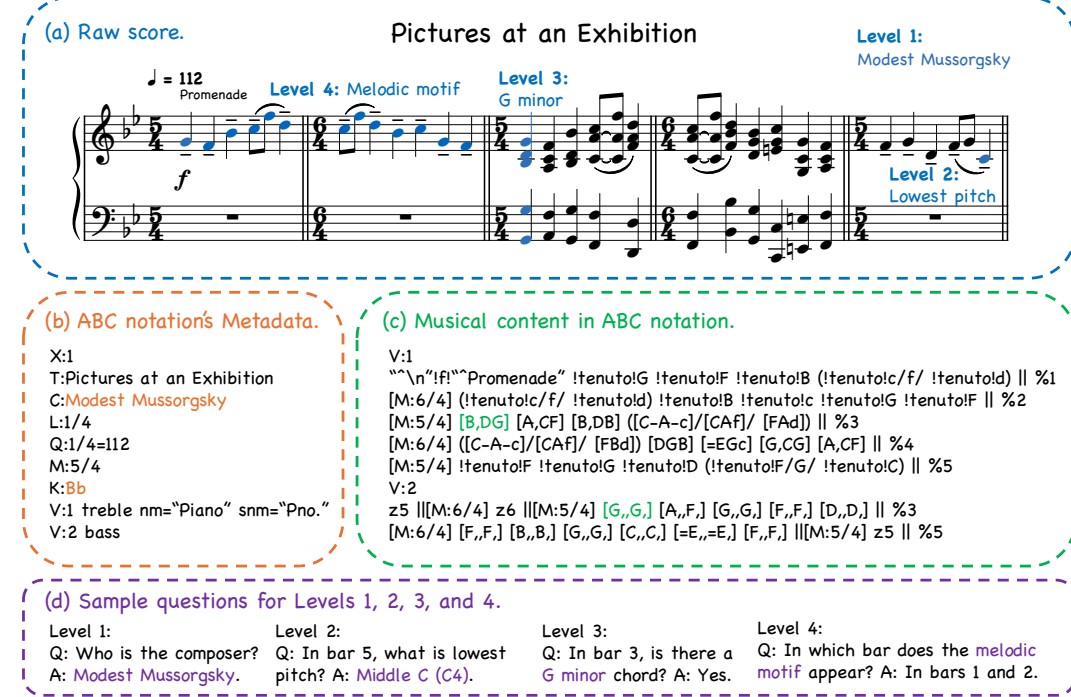

Figure 2: Illustration of multi-level score understanding in MSU-Bench using Mussorgsky's Pictures at an Exhibition. (a) Raw score excerpt with annotated tasks across four levels of difficulty. (b) Metadata encoded in ABC notation. (c) Musical content represented in ABC notation, including voices and chord structures. (d) Sample questions for each level, demonstrating progression from foundational concepts to higher-level musical reasoning.

composer is indicated with `C:Modest Mussorgsky`. The default note length is set with `L:1/4`, meaning that a quarter note is the basic rhythmic unit. The tempo is specified by `Q:1/4=112`. The time signature is written as `M:5/4`, establishing a five-beat measure, though this changes later in the music. The key is marked as `K:Bb`, placing the piece in B-flat major. Finally, `V:1 treble nm="Piano" snm="Pno."` assigns the first voice to the treble clef, labelled as "Piano" (with the short form "Pno."), and `V:2 bass` assigns the second voice to the bass clef.

**Musical Content.** The first voice `V1` corresponds to the right-hand part of the piano. It begins with the annotation "Promenade", marked with the dynamic indication `!f!` (forte) and `!tenuto!` articulations. The melodic line includes notes such as `G`, `F`, and `B`, as well as grouped figures like `(c/f/d)`, each separated by double barlines at the conclusion of bars. Within the progression, the time signature alternates between `5/4` and `6/4`, indicated by `[M:5/4]` and `[M:6/4]`, respectively. Chords appear in brackets, such as `[B,DG]` or `[C-A-c]`, to indicate simultaneous pitches. Accidentals are specified explicitly, for example `=E` for E-natural, and each bar is numbered with comments including `%1`, `%2`, and others in sequence. The second voice `V2` provides the left-hand accompaniment in the bass clef. It begins primarily with rests, such as `z5` and `z6`, which denote whole-bar rests of five and six beats, respectively. As the section progresses, low chords are introduced, notated with double commas (`[G,,G,]`), which indicate very low octave placement.

## 3.3 DATA CURATION

The data collection process for MSU-Bench commences with the selection of 150 scores from MuseScore. When a score contains multiple movements, only the first movement is retained. Scores exceeding 300 bars are truncated, without compromising the validity of the questions. The complete list of scores included in MSU-Bench is provided in Appendix A. For visual QA, the PDF of each score is employed, whereas for textual QA, the corresponding MXL file on MuseScore is converted into ABC notation. A comprehensive set of general questions is then developed and categorised into

Table 1: **Comparison of music-related QA benchmarks across multiple dimensions**. A checkmark (✓) indicates the presence of a feature, a cross (✗) denotes its absence, and a triangle (△) represents partial coverage. "MCQs" refers to benchmarks using a multiple-choice question format.

| Dataset | Modality | | Sheet Music QA | Trainable | Homophony | QA Type | Quantity | Source |
|---|---|---|---|---|---|---|---|---|
| | Textual | Visual | | | | | | |
| MMMU (Yue et al., 2024a) | ✗ | ✓ | ✓ | ✗ | ✓ | MCQs | 369 | Web |
| MusiXQA (Chen et al., 2025) | ✗ | ✓ | ✗ | ✓ | ✓ | Generative | 1.3M | Synthetic |
| ZIQI-Eval (Li et al., 2024) | ✓ | ✗ | ✗ | ✗ | ✓ | MCQs | 14244 | LLMs |
| MusicTheoryBench (Yuan et al., 2024) | ✓ | ✗ | △ | ✗ | ✗ | MCQs | 372 | Human-labelled |
| SSMR-Bench (Wang et al., 2025b) | ✓ | ✓ | ✗ | ✓ | ✗ | MCQs | 3200 | Synthetic |
| Ours | ✓ | ✓ | ✓ | ✓ | ✓ | Generative | 1800 | Human-labelled |

three levels of difficulty (Levels 1–3), designed to evaluate a broad range of musical concepts encompassing fundamental notational knowledge. In addition, score-specific questions are designed as Level 4 questions. Representative examples of these questions are provided in Appendix B to illustrate the structure of MSU-Bench. With the exception of Level 1, Levels 2–4 are intentionally designed to include bar localisation tasks, after which corresponding questions are formulated for each RQ identified in Section 3.1.

Questions from Levels 1–3 are defined as general questions, since they can be applied to any score. These questions address topics including notational onset information, pitch analysis, and harmonic relationships, thereby serving as a foundation for evaluating a model's capacity to process and interpret musical scores with increasing complexity. Once this general question set is finalised, each score in MSU-Bench is assigned nine questions in total: three from Level 1, three from Level 2, and three from Level 3. This systematic allocation ensures that every score is evaluated across multiple domains, thus establishing a balanced benchmark for model assessment.

Level 4 comprises score-specific questions that assess the model's ability to interpret more sophisticated musical phenomena, including melodic motifs, thematic development, textural variation, and orchestration. These questions differ across scores and are designed to evaluate the model's sensitivity to musical subtleties that have been largely neglected in previous benchmarks. Each score is assigned three Level 4 questions, resulting in a total of twelve questions per score and an overall benchmark of 1,800 questions.

Finally, reference answers are manually prepared for each question. This procedure guarantees that MSU-Bench is anchored in accurate and rigorously validated annotations. Each answer is carefully reviewed for correctness and completeness, and explicitly aligned with the musical content of the corresponding score.

## 4 Benchmark Analysis

We provide a comprehensive analysis of MSU-Bench, detailing its novelty, the distribution of questions across different levels, and the characteristics of the questions.

As shown in Table 1, MSU-Bench is the first benchmark to assess LLMs and VLLMs on complete musical scores, spanning tasks from basic notation to advanced analysis. Existing benchmarks contribute complementary perspectives: MMMU (369 web-derived MCQs) and ZIQI-Eval (14,244 LLM-generated MCQs) emphasise multiple-choice breadth; MusicTheoryBench (372 human-annotated MCQs) offers curated content with partial sheet-music support; MusiXQA scales to 1.3M synthetic generative questions; and SSMR-Bench (3,200 synthetic MCQs) explores symbolic tasks. MSU-Bench complements these efforts by integrating textual and visual modalities, supporting model trainability, and addressing homophony in full scores, a dimension often overlooked. With 1,800 human-curated generative QA pairs, it balances reliable annotation with open-ended evaluation, aligning with contemporary LLM and VLLM research.

Figure 3 illustrates a balanced design in which each of the four levels accounts for 25% of MSU-Bench. Related question types are consolidated into broader categories, such as the grouping of expression markings with dynamic markings. More details on question types are in Appendix C.

**Level 1** emphasises performance and metadata, with onset information forming the largest proportion, complemented by smaller contributions from composer, title, tempo, and anacrusis.

**Level 2** addresses markings (14%) and symbolic literacy, with note-level features (6%), time values (3%), and ornaments (2%), and key change for modulation comprising 1%.

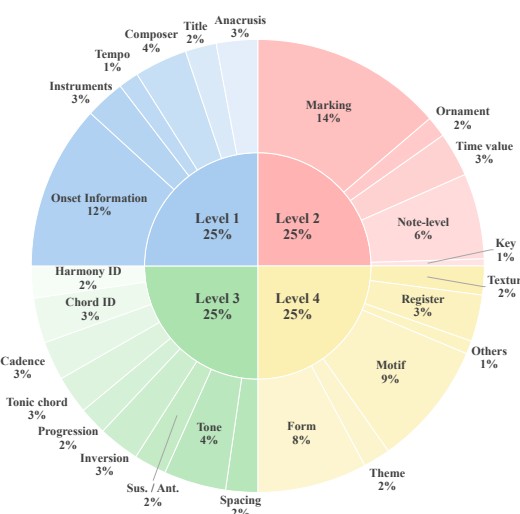

Figure 3: Distribution of 4-Level Questions.

**Level 3** distributes emphasis evenly across harmonic features, including chord identification (ID), cadences, tonic chords, and chord inversions (each 3%), with progressions, suspensions (sus.), anticipations (ant.), chord spacing, and harmonic identification (ID).

**Level 4** highlights broader structural dimensions, with motif (9%) and form (8%) most prominent, supplemented by texture, register, tone, and other questions.

In addition, MSU-Bench encompasses a wide range of composers, as shown in Figure 5, Appendix D, spanning historical periods and stylistic traditions including the Baroque, Classical, Romantic, and twentieth-century repertoire. The distributions of scores by period and genre are presented in Figures 6a and 6b of Appendix D. Collectively, this section highlights the diversity and representativeness of MSU-Bench across major musical dimensions.

## 5 EXPERIMENTS

### 5.1 EXPERIMENT SETTINGS

**Evaluation.** Model outputs are evaluated through a voting process involving ChatGPT-5 (OpenAI, b), Claude Sonnet 4 (Anthropic), and Gemini 2.5 Pro (Google). Accuracy is reported at both the individual level and the aggregate level (overall). We consider two evaluations: (1) **zero-shot**, testing models directly on the 1,800 QA pairs; and (2) **fine-tuned**, where models are adapted with LoRA. We also introduce the **Level-wise Success Rate (LSR)**, capturing the probability of correctly answering successive levels for each score. Let $n$ denote the maximum level, and let $l \in \{1, 2, \ldots, n\}$ be the level index. Then, the LSR at Level $l$ is defined as

$$LSR(l) \ = \ \frac{\text{Correct}(\mathcal{Q}_{1:l})}{|\mathcal{Q}_{1:l}|},$$

where $\mathcal{Q}_l$ denotes the set of all questions belonging to Level $l$. $\mathcal{Q}_{1:l} = \bigcup_{j=1}^{l} \mathcal{Q}_j$ represents the set of all questions from Level 1 through $l$. Correct$(\mathcal{Q}_{1:l})$ indicates the number of instances in which *all* questions from Level 1 through $l$ are answered correctly. $|\mathcal{Q}_{1:l}|$ denotes the total number of questions from Level 1 through $l$. Then, we use the Wilson score interval (Wilson, 1927) to calculate the 95% Confidence Interval (CI) for the LSR at Level $l$, which is given by

$$\hat{p}_l \ = \ LSR(l), \ \text{CI}(l) \ = \ \frac{\hat{p}_l + \frac{z^2}{2n_l}}{1 + \frac{z^2}{n_l}} \ \pm \ \frac{z}{1 + \frac{z^2}{n_l}} \sqrt{\frac{\hat{p}_l(1 - \hat{p}_l)}{n_l} + \frac{z^2}{4n_l^2}},$$

where $\hat{p}_l$ is the LSR at level $l$, and $z$ is the standard normal quantile ($z = 1.96$ for 95% CI).

**Baselines.** We evaluate a diverse set of models for the zero-shot evaluation, including both LLMs and VLLMs. For textual QA (ABC notation), we evaluate ChatGPT-5, ChatGPT-5-mini (OpenAI, b), Claude Opus 4 (Anthropic), Claude Sonnet 4, Deepseek-V3 (DeepSeek-AI), Gemini 2.5 Flash (Google), Gemini 2.5 Pro, Grok 4 (xAI, 2025), Llama 4 Maverick (Meta AI), Llama 4 Scout (Meta AI), Qwen2.5-VL-3B-Instruct (Qwen Team, a), Qwen2.5-VL-32B-Instruct (Qwen Team, a), Qwen2.5-VL-72B-Instruct (Qwen Team, a), Qwen3-4B (Qwen Team, c), Qwen3-32B (Qwen Team, c), Qwen3-Max (Qwen Team, c), and Qwen3-VL-235B-A22B-Instruct (Qwen Team, b).

Table 2: Zero-shot evaluation results on MSU-Bench, with the highest accuracy in **bold**. We evaluate 12 questions per score in a single run for each model to report the accuracy for each level and overall.

| Models | Musical Score Understanding Benchmark | | | | |
|---|---|---|---|---|---|
| | Level 1 (450) | Level 2 (450) | Level 3 (450) | Level 4 (450) | Overall (1800) |
| *Textual QA* | | | | | |
| Qwen3-4B | 20.00 | 10.00 | 08.67 | 13.11 | 12.94 |
| Qwen2.5-VL-3B-Instruct | 32.00 | 09.11 | 17.56 | 13.33 | 18.00 |
| Qwen2.5-VL-72B-Instruct | 34.44 | 18.00 | 18.89 | 12.67 | 21.00 |
| Llama 4 Scout | 48.44 | 25.78 | 26.89 | 26.44 | 31.89 |
| Qwen2.5-VL-32B-Instruct | 50.67 | 20.22 | 26.22 | 37.56 | 33.67 |
| Gemini 2.5 Flash | 50.22 | 31.11 | 30.67 | 24.89 | 34.22 |
| Qwen3-Next-80B-A3B-Instruct | 57.11 | 23.11 | 25.33 | 34.00 | 34.89 |
| Deepseek-V3 | 52.89 | 32.67 | 30.22 | 29.56 | 36.33 |
| Llama 4 Maverick | 52.67 | 31.56 | 28.44 | 33.56 | 36.56 |
| Qwen3-Max | 54.67 | 31.56 | 31.78 | 40.67 | 39.67 |
| Qwen3-VL-235B-A22B-Instruct | 58.44 | 33.56 | 34.00 | 38.89 | 41.22 |
| Claude Opus 4 | 57.11 | 36.89 | 35.56 | 35.56 | 41.28 |
| Claude Sonnet 4 | 61.11 | 40.67 | 35.56 | 33.11 | 42.61 |
| Grok 4 | 62.00 | 40.00 | 31.11 | 37.11 | 42.61 |
| ChatGPT-5-mini | 59.11 | 43.56 | 31.33 | **40.89** | 43.72 |
| ChatGPT-5 | 62.00 | 50.22 | 38.44 | 38.44 | 47.28 |
| Gemini 2.5 Pro | **65.33** | **56.00** | **38.67** | 37.78 | **49.44** |
| *Visual QA* | | | | | |
| Qwen2.5-VL-3B-Instruct | 00.00 | 00.00 | 00.00 | 00.00 | 00.00 |
| Qwen2.5-VL-32B-Instruct | 00.22 | 00.22 | 01.11 | 01.11 | 00.67 |
| ChatGPT-5-mini | 07.11 | 06.67 | 08.89 | 06.67 | 07.33 |
| Grok 4 | 14.00 | 11.11 | 18.44 | 21.33 | 16.22 |
| Qwen3-VL-235B-A22B-Instruct | 18.67 | 15.33 | 22.44 | **22.67** | 19.78 |
| Gemini 2.5 Flash | 19.56 | 15.33 | 29.56 | 18.00 | 20.61 |
| Qwen2.5-VL-72B-Instruct | 21.78 | 18.22 | 27.33 | 18.89 | 21.56 |
| Claude Sonnet 4 | **27.11** | 16.44 | 27.33 | 18.44 | 22.33 |
| Gemini 2.5 Pro | 22.00 | **22.44** | 29.11 | 20.00 | 23.39 |
| Claude Opus 4 | 25.33 | 21.78 | **30.44** | 19.33 | **24.22** |

For visual QA (PDF documents), we include Claude Opus 4, Claude Sonnet 4, Gemini 2.5 Flash, Gemini 2.5 Pro, GPT-5-mini, Grok 4, Qwen2.5-VL-3B-Instruct, Qwen2.5-VL-32B-Instruct, Qwen2.5-VL-72B-Instruct, and Qwen3-VL-235B-A22B-Instruct.

**Models.** We employ Qwen3-0.6B (Qwen Team, a), Qwen3-1.7B (Qwen Team, a), Qwen3-4B, and Qwen2.5-VL-3B-Instruct for the fine-tuned evaluation, adapted using LoRA.

**Data Splitting.** MSU-Bench consists of 150 musical scores. It is divided into training, validation, and testing sets in a 6:2:2 ratio, corresponding to 90, 30, and 30 pieces, respectively. For the fine-tuned evaluation, the testing set's musical scores are extracted from the zero-shot evaluation.

**Training.** We fine-tune the models for 20 epochs on $6\times$A800 GPUs using LoRA (rank 8). Training uses AdamW (Loshchilov & Hutter, 2019) with a $5 \times 10^{-5}$ learning rate, cosine schedule, 10% warm-up, batch size 1, and gradient accumulation of 16. For Qwen2.5-VL-3B-Instruct, we consider three types of input: PDF only, ABC notation only, and their combination (detailed in Appendix E).

## 5.2 EMPIRICAL RESULTS

**Zero-shot Evaluation.** In Table 2, models perform substantially better on the textual QA setting than on the visual QA setting. In textual QA, Gemini 2.5 Pro achieves the best overall accuracy (49.44%), excelling particularly at Level 1 (65.33%) and Level 2 (56.00%). ChatGPT-5 follows closely (47.28%), demonstrating strong stability on higher-level questions (Levels 3–4). Notably, ChatGPT-5-mini attains the highest accuracy on Level 4 (40.89%), suggesting an advantage in more complex reasoning despite its smaller size. Claude Opus 4, Claude Sonnet 4, Grok 4, and Qwen3-VL-235B-A22B-Instruct reach comparable performance (approximately 41.93%), while models

Table 3: Performance of baseline and LoRA-adapted models on MSU-Bench. Qwen2.5-VL-3B-Instruct is adapted using LoRA across the three input modalities outlined in Section 5.1.

| Models | Musical Score Understanding Benchmark | | | | |
| --- | --- | --- | --- | --- | --- |
| | Level 1 | Level 2 | Level 3 | Level 4 | Overall |
| *Textual QA* | | | | | |
| Qwen3-0.6B | 26.67 | 03.33 | 07.78 | 11.11 | 12.22 |
| + LoRA | $55.56^{(+28.89)}$ | $21.11^{(+17.78)}$ | $34.44^{(+26.66)}$ | $37.78^{(+26.67)}$ | $37.22^{(+25.00)}$ |
| Qwen3-1.7B | 30.00 | 10.00 | 01.11 | 18.89 | 15.00 |
| + LoRA | $55.56^{(+25.56)}$ | $24.44^{(+14.44)}$ | $31.11^{(+30.00)}$ | $34.44^{(+15.55)}$ | $36.38^{(+21.38)}$ |
| Qwen3-4B | 47.78 | 17.78 | 06.67 | 20.00 | 23.05 |
| + LoRA | $66.67^{(+18.89)}$ | $38.89^{(+21.11)}$ | $34.44^{(+27.77)}$ | $47.78^{(+27.78)}$ | $46.94^{(+23.89)}$ |
| *Visual QA* | | | | | |
| Qwen2.5-VL-3B-Instruct | 53.33 | 14.44 | 14.44 | 16.67 | 24.72 |
| + PDF | $71.11^{(+17.78)}$ | $33.33^{(+18.89)}$ | $51.11^{(+36.67)}$ | $51.11^{(+34.44)}$ | $50.00^{(+25.28)}$ |
| Qwen2.5-VL-3B-Instruct | 44.44 | 07.78 | 12.22 | 10.00 | 18.61 |
| + ABC | $64.44^{(+20.00)}$ | $34.44^{(+26.66)}$ | $38.89^{(+26.67)}$ | $43.33^{(+33.33)}$ | $45.28^{(+26.67)}$ |
| Qwen2.5-VL-3B-Instruct | 52.22 | 18.89 | 11.11 | 19.10 | 25.34 |
| + PDF&ABC | $68.89^{(+16.67)}$ | $37.78^{(+18.89)}$ | $41.11^{(+30.00)}$ | $48.89^{(+29.79)}$ | $49.17^{(+23.83)}$ |

such as Qwen3-Max and Llama 4 Maverick remain below 40%. Among the open-source models evaluated, Qwen3-VL-235B-A22B-Instruct demonstrates the strongest overall performance, exceeding the text-only Qwen3-Max by about 4%. In contrast, smaller models such as Qwen3-4B and Qwen2.5-VL-3B-Instruct perform considerably worse, thereby highlighting the limitations of lightweight architectures in zero-shot musicological reasoning tasks. The evaluation times of models achieving more than 40% overall accuracy are reported in Appendix F (see Figure 7). While models such as Gemini 2.5 Pro, ChatGPT-5, and ChatGPT-5-mini achieve the highest levels of accuracy, their evaluation times are substantially longer (more than 11 hours). Notably, Qwen3-VL-235B-A22B-Instruct requires only approximately one hour to achieve an overall accuracy of 41.22%.

For visual QA, overall accuracies are markedly lower, with the strongest model (Claude Opus 4) reaching only 24.22%. Claude Opus 4 achieves the highest Level 3 accuracy (30.44%), while Claude Sonnet 4 leads at Level 1 (27.11%) and Gemini 2.5 Pro at Level 2 (22.44%). Most models fail to exceed 20% overall, and smaller variants such as Qwen2.5-VL-3B-Instruct collapse entirely (0.00%). These results highlight a clear modality gap: ABC notation provides a much more reliable representation for large models than raw score images, where recognition and localisation errors continue to dominate performance.

**Fine-tuned Evaluation.** We train the models with LoRA on a question-by-question basis due to the computational constraints imposed by the GPUs. However, we observe that the models achieve substantially better performance when the 12 questions of a score are asked separately. This finding contrasts with our initial expectation that presenting all 12 questions together would enable answers from Level 1 to support responses to higher levels in Table 2. Here, we report the results of models evaluated on a question-by-question basis in Table 3. Table 3 shows that LoRA adaptation yields substantial gains across both textual and visual QA. For textual QA, even small models such as Qwen3-0.6B and Qwen3-1.7B, which achieve only 12–15% overall accuracy in the zero-shot setting on testing set, improve to around 36.8% after LoRA. The effect is even more striking in visual QA: Qwen2.5-VL-3B-Instruct reaches 45–50% after LoRA. These results demonstrate that LoRA adaptation not only closes the gap between LLMs and VLLMs but also unlocks strong textual and visual reasoning capabilities that are absent in zero-shot settings on the testing set.

**LSR Analysis.** Figure 4 shows the LSR from Levels 1–4 across all 150 scores in MSU-Bench. LSR declines steeply with depth in both settings. In textual QA (Figure 4a), models perform moderately at Level 1 (25–35%), with Gemini 2.5 Pro slightly ahead of ChatGPT-5 and Grok 4, but drop below 10% by Level 2 and nearly vanish by Level 3. Visual QA (Figure 4b) is worse: models start at 5–10% on Level 1 and collapse almost entirely by Level 2. The "remaining scores" counts confirm

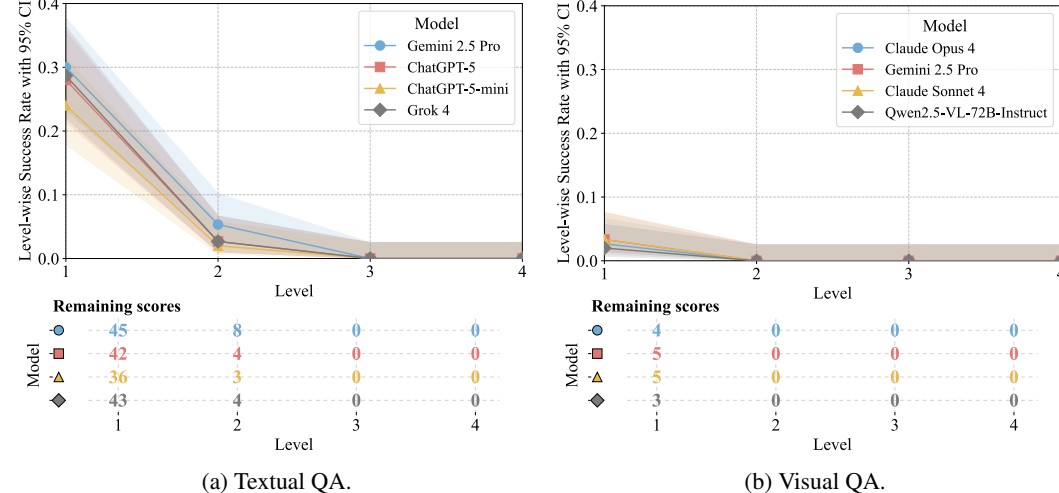

(a) Textual QA.                    (b) Visual QA.

Figure 4: Level-wise Success Rate. We use the entire MSU-Bench to evaluate the performance of various models under textual QA and visual QA. The numbers below each figure indicate the count of scores that remain answerable after each level.

Table 4: Evaluation of models conducted before and after LoRA on MMLU. Qwen2.5-VL-3B-Instruct is adapted using LoRA across the three input modalities described in Section 5.1.

| Models | STEM | Humanities | Social Sciences | Other Subjects |
|---|---|---|---|---|
| Qwen3-4B | 72.63 | 81.44 | 63.21 | 74.61 |
| + LoRA | 74.09$^{(+01.46)}$ | 81.54$^{(+00.10)}$ | 63.51$^{(+00.30)}$ | 75.11$^{(+00.50)}$ |
| Qwen2.5-VL-3B-Instruct | 60.60 | 75.63 | 58.72 | 69.65 |
| + PDF | 60.90$^{(+0.30)}$ | 75.66$^{(+00.03)}$ | 58.45$^{(-00.27)}$ | 69.80$^{(+00.15)}$ |
| + ABC | 60.47$^{(-00.13)}$ | 75.79$^{(+00.16)}$ | 58.13$^{(-00.59)}$ | 69.62$^{(-00.03)}$ |
| + PDF&ABC | 60.50$^{(-00.10)}$ | 75.85$^{(+00.22)}$ | 58.28$^{(-00.44)}$ | 69.65$^{(-00.00)}$ |

this fragility: about 41.5 scores survive past Level 1 in textual QA versus only around 4.25 in visual QA, with nearly all failing by Level 2. These results highlight that while models can solve isolated questions, sustaining correctness across levels is extremely difficult, underscoring LSR's diagnostic strictness. LSR for LoRA-adapted models is reported in Appendix G.

**Massive Multitask Language Understanding (MMLU).** We evaluate the models adapted with LoRA and those without adaptation to assess forgetting on MMLU (Hendrycks et al., 2021). MMLU evaluates models across 57 distinct subjects, encompassing Science, Technology, Engineering, and Mathematics (STEM), as well as the humanities, social sciences, and other subjects such as law and medicine. As shown in Table 4, the models adapted with LoRA exhibit minimal forgetting, with performance remaining close to that of their base versions. These results indicate that LoRA adaptation effectively preserves the models' general knowledge while enhancing their capabilities in musical score understanding and reasoning.

## 6 CONCLUSION AND FUTURE DIRECTIONS

We introduced Musical Score Understanding Benchmark (MSU-Bench), a benchmark that for the first time evaluates LLMs and VLLMs on holistic musical score reasoning across textual (ABC notation) and visual (PDF) modalities. Through comprehensive experiments, we demonstrate that current models struggle to sustain multi-level comprehension, especially under visual QA, but that ABC representations and lightweight adaptation techniques such as LoRA significantly mitigate these challenges. Our findings suggest that effective musical score understanding requires both robust bar localisation and grounding mechanisms, as well as multimodal alignment across textual and visual formats. We envision MSU-Bench as a foundation for future research at the intersection of AI, musicology, and multimodal reasoning, encouraging the development of models that not only read and reason about language but also comprehend the structural richness of music.

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

648
649

# A APPENDIX

650
651

## A.1 LIST OF THE MUSICAL SCORES

652
653
654

1. Cello Suite No.1 BWV 1007 - 1. Prélude

655

2. Solfeggietto in C minor

656

3. Toccata and Fugue in D minor BWV 565

657

4. Fugue in G Minor BWV 542

658
659

5. Fugue I in C major BWV 846

660

6. Fugue in D minor BWV 948

661

7. Fugue in G Minor BWV 578

662
663

8. Prelude I in C major BWV 846

9. Sonate No. 16 1st Movement

664

10. Piano Sonata No. 5 in C Minor Op.10 No.1

665

11. Sonata in G Op.14 No.2 Movement 1

666

12. Piano Sonata in A major Op.2 No.2

667
668
669

13. Piano Sonata No. 3 in C Major Op. 2 No. 3

14. Sonata No. 23 Op. 57 Appassionata

670

15. Sonata Op.31 No.17 in D minor Tempest

671
672

16. Piano Sonata No. 17 in D minor Op. 31 No. 2

673
674

17. Piano Sonata No.18 in E flat major Op.31 No.3

675
676

18. Sonate No.8 Op.13 Pathétique 3 Rondo. Allegro Sonata No.8

677

19. Les Nuits d'été

678

20. Symphonie fantastique, H 48

679

21. Polovtsian Dances

680
681

22. Hungarian Dance No. 5

682

23. Rhapsody Op. 79 No. 2

683

24. Waltz Op.39 No.3

684

25. Intermezzo in E flat major Op.117 No.1

685
686

26. B minor Rhapsody 1 Op. 79

687

27. Ballade Op.118 No.3

688

28. Intermezzo Op. 116 No. 2

689

29. Intermezzo Op. 118 No. 2 A Major

690
691

30. Violin Concerto in E minor Op.64

692

31. Lullaby Op.49 No.4

693

32. Waltz in A Major Op.39 No.15

694

33. Fantaisie-Impromptu in C♯ Minor

695
696

34. Nocturne Op. 9 No.1

697

35. Nocturne-No. 20 in C Sharp Minor

698

36. Ballade no.1 in G minor Op.23

699

37. Sonata No.2 Op.35 1st Movement

700
701

38. Ballade No.3 in A flat major Op.47

39. Ballade No.4 in F minor Op

40. Prélude Opus 28 No. 4 in E Minor

41. Waltz in A Minor

42. Nocturne Op. 27, No. 2

43. Suite Bergamasque

44. La fille aux cheveux de lin

45. Reverie

46. Clair de lune

47. Premier Trio

48. Syrinx

49. Sonate pour Violoncelle et Piano

50. Symphony No. 9 New World II, Largo

51. Symphony No. 9 New World:IV, Allegro con fuoco

52. Humoresque No.7

53. Holberg Suite Op.40 I.Praeludium

54. Wedding Day at Troldhaugen

55. Anitras Dance Piano solo

56. Dance Op. 12 No. 4

57. Sailors Song Op.68 No.1

58. Waltz Op. 12 No. 2

59. Butterfly Sommerfugl Op. 43 No. 1

60. Piano Concerto in A minor Op.16

61. In the Hall of the Mountain King

62. Lyric Pieces Op.47 Grieg

63. Lyric Pieces Op. 54 No. 4

64. Morning Mood from Peer Gynt Suite No. 1

65. Sonata in E Minor, Hob. XVI: 34 (I: Presto)

66. String quartet - Op.76, No.5, in D major

67. Cello Concerto C Major Movement 1

68. Piano Sonata in F Major HOB.XVI/23

69. Haydn Sonata Hob. XVI37 Mov. 1 D Major

70. String Quartet Op.64 No.3

71. Piano Concerto in D major

72. Die Schöpfung Mit Würd' und Hoheit angetan

73. Piano Sonata in E minor HOB. XVI/34

74. Sonata in C minor HOB/XVI:20

75. String Quartet in C major ("Emperor") Op. 76 No. 3

76. Die Fledermaus Grunfeld Op. 56 Konzertparaphrase

77. Radetzky March

78. Pizzicato Polka Arranged for Piano Solo

79. The Blue Danube Accordion Solo
80. Tratsch-Polka Op.214
81. Strauss Die Fledermaus Op.362 Overture
82. Hungarian Rhapsody No. 2
83. Etude S.136 No.4
84. Trois Etudes de Concert No. 3
85. Der Müller Und Der Bach. D795, S.5652
86. Hungarian Rhapsody No. 6
87. Etude S.136 No.5
88. Etude S.136 No.9
89. William Tell Overture Finale
90. Romance S.169
91. Grandes études de Paganini, S.141: No. 6
92. S. 1413 in G♯ Minor, La Campanella
93. S.541 No.3 in A♭ Major
94. Symphony No.10 - I. Adagio Complete Score
95. Song Without Words Op.85 No.3
96. Song without Words Op. 38 No.6
97. Song Without Words Op.30 No.5
98. Melodie Op.4 No.2 in C minor
99. Songs without words Op.30 No.1
100. Songs Without Words Op.19 No.6
101. Songs Without Words Op.62
102. Wedding March
103. Songs Without Words Op.19 No.4
104. Song Without Words Op.19b No.1
105. Songs Without Words Op.19 No.3
106. Piano Sonata No.1 K.279
107. Sonata No. 5 1st Movement K.283
108. Sonata No. 7 1st Movement K. 309
109. Piano Sonata No.8 in A minor K.310300d
110. Piano Sonata No. 8 in D Major, K. 311 (284c): I. Allegro con spirito
111. Piano Sonata No.18 in D major K
112. Piano Concerto No.23 in A major K.488
113. Mozart Rondo Alla Turca
114. Piano Sonata No. 16 - Allegro
115. Sonata No.11 in A major K.331
116. Pictures at an Exhibition: No.2, Il vecchio castello
117. Pictures at an Exhibition 13 8. Catacombae
118. Pictures at an Exhibition 14 Cum mortuis in lingua mortua
119. Pictures at an Exhibition Movement 15 (No.9)
120. Pictures at an Exhibition 16 10
121. Pictures at an Exhibition-Gnomus (The Gnome) & Promenade
122. Pictures at an Exhibition
123. Strauss Die Fledermaus Op.410 Overture
124. Piano Concerto in G major - II
125. Gaspard de la Nuit, No.2, "Le Gibet"
126. Gaspard de la Nuit, No. 1, "Ondine"
127. Flight of the Bumblebee Piano
128. Concerto No.1 in a minor
129. Sans The Cuckoo in the Depths of the Woods
130. Sans - Fossils Transcribed for Piano
131. 2nd Piano Concerto 1st Movement Piano solo
132. Introduction and Rondo Capriccioso Op.28
133. Le Cygne The Swan Easy Piano by Free MusicKey
134. Allegro Appassionato Cello Piano
135. Piano Sonata D.784 - 1st movement
136. Impromptu in C minor No.1 Op.90
137. Impromptu No. 3 Op. 90 D 899 G♭ Major Transcription
138. Impromptu No.3 Op.90 D 899 G Majeur Transcription de Liszt
139. Piano Sonata No.19 in C minor
140. Sonata Op.42
141. Ave Maria
142. Winterreise D.911 No.1 - Gute Nacht
143. Winterreise D.911 No.5 - Der Lindenbaum
144. Die Forelle D. 550 Op. 32
145. Winterreise D.911 No.24 - Der Leiermann
146. Schwanengesang D.957 No.4
147. Waltz Op. 18 no. 6 in B minor
148. Piano Sonata No.2 in G
149. Vers La Flamme Op.72
150. Etude Opus 8 No. 12 in D Minor

# B   APPENDIX

## B.1   SAMPLE QUESTIONS

Using *Solfeggietto in C minor* by J.S. Bach as an illustrative example, the following section presents sample questions from each of the four levels in MSU-Bench. In total, the dataset comprises 1,800 QA pairs drawn from the 150 complete musical scores, with 450 questions allocated to each level. The structure of these questions is exemplified below.

**Level 1**

- **Q1:** Is the piece written with an anacrusis (upbeat bar)?
  **A1:** No.
- **Q2:** What is the tempo in beats per minute?
  **A2:** 150 bpm.
- **Q3:** What is the initial key?
  **A3:** C minor.

**Level 2**

- **Q1:** In bar 1, what is the dynamic marking?
  **A1:** f.
- **Q2:** In bar 11, is there an accidental, and what is it?
  **A2:** F♯4, A♮4.
- **Q3:** In bar 25, what is the ornament on note D2?
  **A3:** Passing note, neighbour note.

**Level 3**

- **Q1:** In bar 33, what is the chord progression? (use I, II, III, IV, V, VI, VII)
  **A1:** I–V7.
- **Q2:** In bar 27, what scale degree is the first chord? (use I, II, III, IV, V, VI, VII)
  **A2:** I.
- **Q3:** In bar 12, is there a dominant chord?
  **A3:** No.

**Level 4**

- **Q1:** What is the predominant rhythm of the piece?
  **A1:** Semiquaver.
- **Q2:** In bar 1, in which register is the melody?
  **A2:** Middle register.
- **Q3:** What are the main features of the motif?
  **A3:** Third interval.

# C    APPENDIX

## C.1    DETAILED BREAKDOWN OF QUESTION TYPES

In this section, we provide a comprehensive breakdown of the question types encompassed within each level of MSU-Bench, as illustrated in Figure 3.

### C.1.1    LEVEL 1

**Onset Information.** The questions ask about onset information that appears at the very beginning of a musical score, including the composer and title, the initial key, clef, and time signature, the presence of an anacrusis, the instruments involved, as well as the opening tempo, metronome marking, and expression indications. Collectively, these elements establish the basic identity, notation, and performance instructions that frame how the piece is read and interpreted from the outset.

### C.1.2    LEVEL 2

**Notation & Note.** Level 2 questions differ from Level 1 questions in both scope and depth. Whereas Level 1 focuses on onset information that is immediately visible at the start of a score, such as composer, title, initial key, time signature, clef, instrumentation, tempo, and expression indications. Level 2 shifts attention to localised details within the body of the music. These questions aim to examine specific bars for note-level features (highest or lowest pitch, presence of accidentals, shortest or longest note and rest values), performance instructions (dynamics, articulation, ornaments, tempo changes, expression markings), and structural signs (repeat signs, clef or key changes, modulation).

### C.1.3    LEVEL 3

**Harmony & Chord.** Level 3 questions delve into mid-level musical structures, focusing on harmonic and chordal analysis within specific bars. Level 3 progresses from the recognition of individual symbols to the analytical reasoning required for understanding harmonic structure. Rather than focusing on isolated surface features, such as a dynamic marking or an accidental within a single bar, these questions require the interpretation of tonal function. LLMs or VLLMs are expected to identify chords by scale degree using Roman numerals, determine inversions and spacing, recognise cadences including perfect, imperfect, and interrupted, and distinguish non-chord tones such as suspensions (sus.) and anticipations (ant.). In addition, this level addresses harmonic progressions extending across multiple bars, examines whether passages commence or conclude on the tonic, and consider the treatment of chord tones that are omitted or repeated.

### C.1.4    LEVEL 4

**Texture & Form.** Level 4 extends beyond the recognition of notation and harmony to encompass piece-specific understanding and large-scale structural analysis. The questions require the identification of a work's genre and performance medium, including its orchestration or ensemble, as well as its formal design, such as principal and secondary themes, transitions, and main sections. They also address thematic materials by asking where a motif first appears, its defining features in terms of rhythm, register, or instrumental part, and the ways in which it is developed or ornamented. Further areas of focus include prevailing textural and accompanimental conventions, the predominant tempo and rhythmic character of the work, and, on occasion (others), the number of movements. In the case of instrument-specific repertoire, idiomatic features such as bowings are considered. Overall, Level 4 demands the synthesis of information across extended spans of music in order to characterise style, form, texture, and thematic organisation, thereby moving towards holistic musical analysis rather than bar-by-bar observation.

# D   APPENDIX

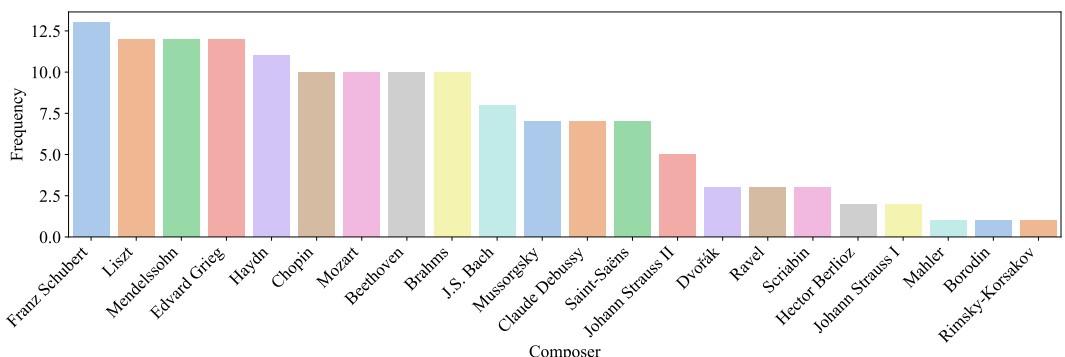

Figure 5: Frequency distribution of composers represented in MSU-Bench. The histogram illustrates the number of pieces per composer, with Franz Schubert, Liszt, Mendelssohn, and Edvard Grieg appearing most frequently, while representation gradually decreases for others.

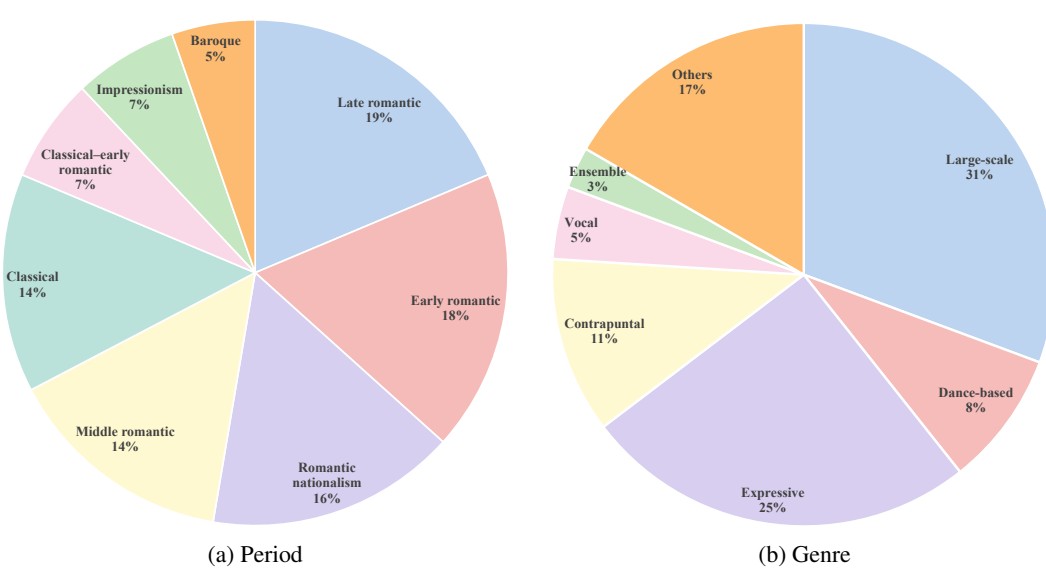

(a) Period                                          (b) Genre

Figure 6: Distribution of musical periods and genres in MSU-Bench. (a) shows the historical periods of the selected scores, ranging from Baroque (5%) to Impressionism (7%) and various stages of Romanticism, with Late Romantic (19%) and Early Romantic (18%) being most prominent. (b) presents the genre distribution, where large-scale works (31%) and expressive pieces (25%) constitute the majority, followed by contrapuntal (11%), dance-based (8%), and other categories.

# E  APPENDIX

## E.1  DIFFERENT TRAINING SETTINGS FOR QWEN2.5-VL-3B-INSTRUCT

To investigate the effect of different input modalities on model adaptation, we design three distinct fine-tuning strategies for Qwen2.5-VL-3B-Instruct.

**PDF.** In this setting, we treat PDF sheet music as the sole input modality. The model receives images rendered from PDF pages, and both the visual encoder and the language model are updated during training. This setting evaluates the model's ability to extract structural and symbolic information directly from visual sheet-music representations.

**ABC notation.** Here, we replace PDF images with ABC notation as the only training input. Since this modality does not require visual parsing, we freeze the visual encoder to reduce computational overhead and update only the language model and LoRA adapters. This strategy evaluates whether ABC notation alone is sufficient for enabling VLLMs to learn music-theoretical patterns.

**PDF + ABC notation.** In the multimodal setting, we provide both PDF images and ABC notation for each score. Both the visual encoder and the language model are fine-tuned. The objective is to examine whether complementary information from visual and symbolic modalities produces better performance than unimodal training. By integrating structural cues from PDF files with explicit symbolic tokens from ABC, this approach is expected to enhance robustness and generalisation across diverse tasks. However, the combination of both modalities constrains the maximum number of tokens available for training. Consequently, the following pieces are excluded:

1. *Piano Sonata No. 5 in C Minor, Op. 10 No. 1*
2. *Sonate pour Violoncelle et Piano*
3. *Piano Concerto in A Minor, Op. 16*
4. *Hungarian Rhapsody No. 2*
5. *Ballade No. 1 in G Minor, Op. 23*
6. *Ballade No. 4 in F Minor, Op. 52*
7. *Sonata, Op. 42*
8. *Concerto No. 1 in A Minor*

# F  APPENDIX

## F.1  EVALUATION TIME

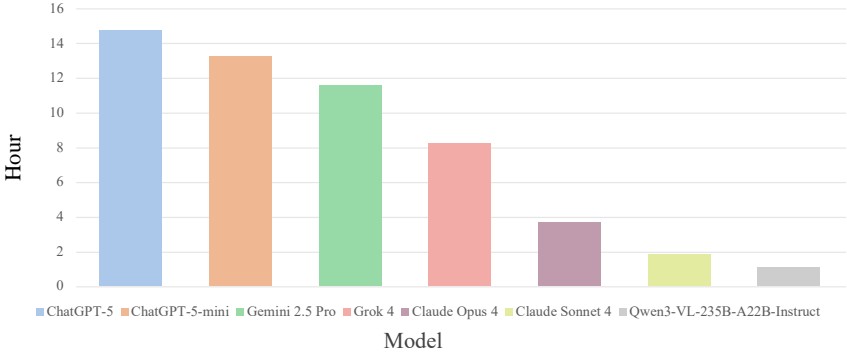

Figure 7: The evaluation time for models exceeding 40% overall accuracy.

# G    APPENDIX

## G.1    LSR FOR MODELS ADAPTED USING LoRA

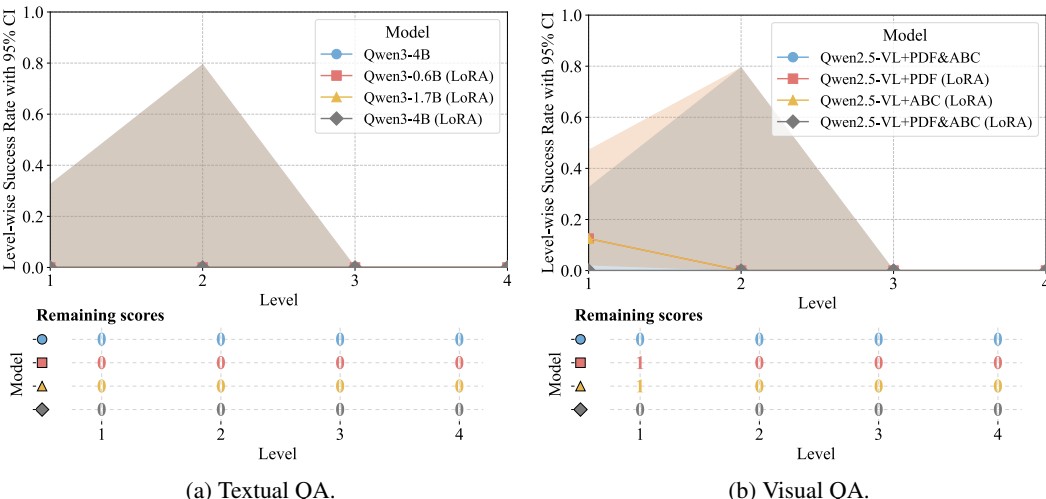

(a) Textual QA.    (b) Visual QA.

Figure 8: Level-wise Success Rate for Models Adapted Using LoRA. We use the testing set held out from MSU-Bench to evaluate the performance of models fine-tuned with LoRA under different input modalities. (a) shows the results for models trained solely on ABC notation, while (b) presents the results for models trained using the three input modalities: PDF only, ABC only, and both PDF and ABC, compared to the baseline Qwen2.5-VL-3B-Instruct, that uses both PDF and ABC as input.

# H    APPENDIX

## H.1    USE OF LLMS

In preparing this manuscript, we employ LLMs such as ChatGPT-5 solely as an auxiliary tool for academic writing. Their use is restricted to *linguistic refinement*, including polishing grammar, improving clarity and fluency, and adjusting the structure and formatting of text and tables. We do not rely on LLMs for generating research ideas, designing methodologies, conducting experiments, performing data analysis, or interpreting results. All conceptual contributions, experimental designs, and substantive findings reported in this work are entirely our own.

