# OpenReview forum: "Musical Score Understanding Benchmark: Evaluating Large Language Models’ Comprehension of Complete Musical Scores"
_ICLR.cc/2026/Conference — ICLR 2026 Conference Withdrawn Submission_

### Official Review · Reviewer_hE7M · 2025-10-26

**Soundness:** 3
**Presentation:** 2
**Contribution:** 2
**Rating:** 4
**Confidence:** 4

**Summary:**

This paper presents MSU-Bench, a large scale benchmark and finetuning set for evaluating LLM and VLLMs on symbolic music reasoning tasks.

**Strengths:**

- The scale in which the authors purportedly constructed the dataset using human labelers is quite strong. I don't think another symbolic music benchmark exists that has this level of human-verified questions and answers.
- In general, overall empirical setup for using the benchmark is sound and bears no clear issues.

**Weaknesses:**

- Overall, the structure and writing of the paper is somewhat poor. The authors consistently restate the definitions of the 4 levels of MSU-Bench, taking up Section 3.1, pars of 3.2, and parts of Section 4. While the clarity is welcome, this muddies the overall readability of the paper and adds a level of redundancy that obfuscates the key contributions of the paper. Additionally, while the case study *figure* is very informative, the text accompanying it is hard to parse, and feels relatively useless in the presence of the figure.
- It would be useful to have more details in terms of how the benchmark was constructed (and with reduction of the above redundancies, there would be space for it). In particular, for level 1-3 questions, were these created by humans by hand, or were they automatically extracted from the ABC format itself (and if so, how)? For reference answer generation, *who* was generating the answers (demographic info, experience with music, etc.), and was there any confirmation from other answer creators that these answers were correct?
- The point made in the intro regarding VLLMs challenges on score reading, and ABC and MSU-Bench as a proposed solution, are not fully sound. Implicitly stating that the process of visual score->ABC as some consistently performable operation is misrepresented; the large majority of score images online do not have corresponding MXL/XML files to easily parse, and thus this argument only holds in the exclusive case of musescore / xml data. Additionally, there is nothing inherent the analysis of MSU-Bench that confirms assessment of localization: just because there are questions that are about localized content does not mean the evaluation LLMs/VLLMs are actually *using* this localized content to make their answer.
- To solve this previous point, it would be useful if the authors followed the analysis of recent previous works [1,2,3] that broadly assess performance with AND without the multimodal conditions for the same model (i.e. asking questions with no image / random noise or no ABC). This would more accurately target whether the models are *guessing* about localized content or actually picking on these local cues.
- There is no verification that the voting process used for evaluation is sound in any way. While I understand the authors' goals of moving away from MCQ-type benchmarks, to propose a benchmark with full LLM-as-a-judge evaluation requires at least some verifications that the outputs of the voting process (which are also not detailed in the paper) actually correlate with human perception of "correctness".
- Homophony (Table 1) is never defined within the paper.
- Missing citation for [3] in modern sheet music-LLM evaluation benchmarks.

Overall, while I think the benchmark does have value (especially if it is all fully human created), there are too many issues with the manuscript in its current form for me to recommend acceptance. Changes to these would significantly improve the quality of the paper and I would adjust my score accordingly.


[1] Kumar, Sonal, et al. "Mmau-pro: A challenging and comprehensive benchmark for holistic evaluation of audio general intelligence." arXiv preprint arXiv:2508.13992 (2025).

[2] Zang, Yongyi, et al. "Are you really listening? boosting perceptual awareness in music-qa benchmarks." ISMIR (2025).

[3] Mundada, Gagan, et al. "WildScore: Benchmarking MLLMs in-the-Wild Symbolic Music Reasoning." EMNLP (2025).

**Questions:**

See weaknesses.

---

### Official Review · Reviewer_kC54 · 2025-10-28

**Soundness:** 3
**Presentation:** 3
**Contribution:** 2
**Rating:** 4
**Confidence:** 5

**Summary:**

This paper introduces MSU-Bench, a dataset for evaluating how large language and vision-language models understand complete musical scores. It contains 150 MuseScore pieces, each available as both PDF (visual) and ABC notation (text), with 1,800 manually written question–answer pairs across four levels of musical comprehension—from basic metadata to harmony and form. The authors benchmark over 15 recent models and test LoRA fine-tuning on a subset of scores, showing that models perform much better on textual notation than on visual scores. While LoRA improves accuracy without harming general knowledge, the dataset remains small and primarily serves as a controlled benchmark rather than a training resource.

**Strengths:**

Strengths:

The paper designs and evaluates musical comprehension of VLLMs.
1. Dual-modality setup with ABC notation and PDF formats.
2. Human-written, theory-based questions with verified answers.
3. Comprehensive evaluation across many recent LLMs and VLLMs.
4. Introduction of the Level-wise Success Rate (LSR) metric for hierarchical reasoning.

**Weaknesses:**

The dataset is far too limited in scale and representational diversity to reflect the complexity of musical notation. Musical notation systems (especially Western classical) include a large vocabulary of distinct symbols, easily numbering in the hundreds to low thousands when counting both core and auxiliary marks. Their combinations across pitch, rhythm, articulation, and dynamics create enormous structural variety. With only 150 MuseScore-rendered PDFs and 1,800 QA pairs, the benchmark cannot possibly cover enough variation or symbol combinations for a VLLM to learn decomposed, structured understanding of music scores. The short, categorical answers further restrict the task to local feature recognition rather than true notation reading or transcription.
1. Severe scale limitation: Hundreds to thousands of notation symbols and vast combinational patterns exist, but only 150 PDFs are included—insufficient for robust multimodal learning.
2. Shallow answer format: Responses are short text labels (e.g., key, chord, bar index) rather than symbolic or sequential transcriptions.
3. No real score reading: Models identify local cues but are not tested on end-to-end notation parsing or structural understanding.
4. Uniform visual data: MuseScore engravings lack stylistic or handwritten variation found in real-world scores.
5. Limited generalization: The dataset’s narrow scope prevents models from learning or evaluating decomposed representations of complex musical notation.

**Questions:**

Questions and Suggestions for the Authors:
1. Data diversity: The benchmark uses 150 MuseScore-rendered PDFs. Could the authors clarify whether these are all digitally engraved, or if any scanned or handwritten scores were included to represent real-world notation variance?
2. Symbol coverage: Musical notation contains hundreds to thousands of symbols and combinations. How do the authors assess whether the current dataset covers enough symbolic variety for meaningful model evaluation?
3. Answer format: Since answers are short text labels rather than note sequences, the benchmark mainly tests local recognition. Would the authors consider adding tasks that involve full-score parsing or symbolic transcription similar to OMR?

---

### Official Review · Reviewer_FJS7 · 2025-10-31

**Soundness:** 2
**Presentation:** 3
**Contribution:** 2
**Rating:** 2
**Confidence:** 4

**Summary:**

This paper proposes MSU-Bench, a new benchmark designed to evaluate the ability of Large Language Models (LLMs) and Vision-Language Models (VLMs) to understand complete musical scores. The benchmark contains 1,800 human-curated generative QA pairs derived from 150–200 Western classical scores, presented in both ABC notation (text) and PDF score images (vision). The questions are organised into four hierarchical levels of musical comprehension

**Strengths:**

Originality & Relevance.
The paper introduces MSU-Bench, a new benchmark targeting full musical score understanding across both symbolic (ABC notation) and visual (PDF) modalities. This is a timely and underexplored direction, extending multimodal LLM evaluation beyond language, images, and speech into symbolic music reasoning.

Technical Contribution.
The benchmark includes 1,800 human-annotated QA pairs spanning multiple hierarchical levels of score comprehension, enabling fine-grained analysis of model failure modes. The comparison of zero-shot vs. LoRA-tuned models provides useful insight into adaptation and multimodal gaps.

Significance for the Community.
The work can catalyze research at the intersection of LLMs, musicology, and symbolic reasoning. The dataset, evaluation protocol, and findings are likely to be reused by future work on music reasoning, OMR, or symbolic multimodal learning.

**Weaknesses:**

Scope of Musical Knowledge is Narrow.
The benchmark focuses almost exclusively on Western classical repertoire and a limited subset of forms (mostly first movements, 150–200 scores). It does not include non-Western traditions (e.g., East Asian, Middle Eastern maqam, African percussion, Latin American music, etc.), handwritten scores, or modern/medieval works. This limits its claim of “complete score understanding” and introduces cultural and stylistic bias.

Limited Musicological Depth in Question Design.
The benchmark mainly targets symbolic and structural reasoning (pitch, rhythm, harmony, form). It omits equally important musicological dimensions that LLMs could answer from domain knowledge, such as performance practice, emotional response, historical context, instrument techniques, or difficulty level—topics central to real-world music analysis.

Potential Dataset Leakage & Training Ambiguity.
The paper does not clearly explain the mapping between the 150 scores and the 1,800 QA pairs used in LoRA training. It is unclear whether test questions overlap with adapted data, raising concerns about leakage and generalisation claims.

Lack of Discussion on Language Bias in Multiple Choice Options.
Since the benchmark uses four-option questions, it is unclear whether models exploit poorly-designed distractors rather than true score understanding. There is no ablation that tests whether models can guess the correct answer without ABC/PDF input.

Statistical Analysis of Results Is Weak.
Several reported model comparisons rely on very small absolute differences (e.g., +1.55%, +10 correct answers out of 450) without statistical significance testing. Claims such as “smaller model performs better on Level 4 reasoning” are not backed by variance analysis or hypothesis testing.

Baseline Performance Below Random Guessing.
Some VLM accuracies fall below the random-guess baseline of 25%, which is not discussed. It is unclear whether models fail due to instruction-following problems, poor visual parsing, or systematic bias in distractor design.

And it lack of quality evaluation, such as report the accuracy of human experts (musicians)

**Questions:**

Dataset Scope & Bias
Do you plan to expand beyond Western classical scores, handwritten notation, and first-movement bias? If not, how should results be interpreted with respect to general symbolic music reasoning?

Coverage of Musicological Tasks
Why were only structural/symbolic questions included? Could the benchmark be extended to include performance difficulty, emotional response, historical context, etc., which LLMs are capable of answering?

LoRA Training Transparency
Please clarify the exact split between training and evaluation scores. Were any QA pairs from the 150 LoRA-adapted scores included in the test set? If so, how do you ensure no leakage?

Language Bias in Answer Choices
Have you tested whether models can guess answers without reading the score (i.e., without ABC/PDF input)? This would help quantify distractor bias.

Statistical Validity of Reported Gains
Could you report significance testing for small gains such as +1–2% accuracy? Are the Level-4 improvements of smaller models statistically meaningful or random?

Below-Random VLM Accuracy
How do you explain models scoring <25%? Are they failing to follow instructions, or systematically hallucinating due to visual bar mislocalisation?

Generalisation of LoRA Beyond MSU-Bench
Does the LoRA adaptation transfer to other OMR datasets?

**Details Of Ethics Concerns:**

copyright of music score need to be discussed.

---

### Official Review · Reviewer_PqHT · 2025-11-04

**Soundness:** 3
**Presentation:** 3
**Contribution:** 2
**Rating:** 4
**Confidence:** 3

**Summary:**

The paper presents a human-curated benchmark for musical score understanding in both textual and visual modalities. The authors introduced four levels of musical understanding questions, and evaluated many LLMs and VLLMs in both zero-shot and finetuned settings.

**Strengths:**

- The paper releases a human-curated benchmark of 1800 QA pairs across 150 full musical scores, which is a useful resource for the community.
- It organizes questions into four levels (from local notation to harmony and form), which is an intuitive way to structure musical understanding.
- The dataset is multimodal (ABC text and PDF images), and the authors run broad benchmarks on both LLMs and VLMs and include LoRA fine-tuning results, showing the dataset can drive measurable gains.
- The full score setting targets more realistic, long-context scenarios that many current systems struggle with.

**Weaknesses:**

1. To me the novelty of this paper appears to be incremental: this is primarily a dataset evaluation rather than a data curation method and similar human-curated music resources exist.
2. Although the authors introduced four levels of musical understanding in a progressive manner, they mostly function as independent categories in finetuning and evaluation.
3. The evaluation relies on LLM voting for judging open-ended answers. This is fragile and there’s no human-calibration study to validate the judges.
4. The fine-tuning evidence is weak. The data only has 1800 QA pairs and there is no result showing if the improvement also transfer to other music benchmarks, which makes the broader impact is unclear.
5. The paper motivates with “localization and hallucination” in visual models, but experiments don’t separate perception (OMR/layout) errors from reasoning—e.g., no condition that feeds gold ABC to the visual models or uses bar-level crops to reduce layout noise.

**Questions:**

1. Can the authors explain the selection of the datasets and provide more details whether different period/genres may affect the LLMs and VLLMs performances?
2. What exact prompts and temperatures were used for LLM voting, and how stable are results across prompt variants or judge models?
3. Can the authors provide more details of finetuning experiments. For VLMs, which modules were updated (vision encoder, projector, language backbone)?

---

### Note · Authors · 2025-11-28

I have read and agree with the venue's withdrawal policy on behalf of myself and my co-authors.